# Research on the Structural Characteristics and Evolutionary Process of China’s Tourism Investment Spatial Correlation Network

**DOI:** 10.3390/ijerph192315661

**Published:** 2022-11-25

**Authors:** Haijian Li, Wujie Xie

**Affiliations:** School of History Culture and Tourism, Jiangsu Normal University, Xuzhou 221116, China

**Keywords:** tourism investment, space structure, social network, China

## Abstract

The paper uses the revised gravity model to measure the intensity of tourism investment spatial correlation, constructs the spatial correlation matrix of tourism investment, and uses the social network method to analyze the structural characteristics and evolutionary process of tourism investment spatial correlation network based on 31 provinces in China from 2000 to 2016. The findings revealed: (1) The spatial correlation quantity of interprovincial tourism investment continues to grow, with Beijing, Jiangsu, Zhejiang, Shanghai, Shandong, and Guangdong at the top of the list. (2) Overall network density and correlation are rising, and the spatial correlation of interprovincial tourism investment is increasingly close. Network hierarchy and network efficiency are decreasing, and network stability has been enhanced. (3) Degree centrality and closeness centrality of each province have shown a significant increase; Beijing, Shandong, Guangdong, Jiangsu, Zhejiang, and Shanghai are the top six and in the center of the network. Most provinces have improved betweenness centrality, Beijing, Guangdong, Shandong, Liaoning, Shaanxi, and Hunan have a strong betweenness centrality with strong intermediary capacity. (4) The core area mainly includes eastern and central provinces, and the periphery areas mainly include western and northeastern provinces. The network connection density of the core and periphery areas shows an increasing trend, while the network linkage density between the core and periphery areas shows a downward trend.

## 1. Introduction

Tourism investment is one of the most important sources of economic development, providing an important guarantee for tourism to expand production, promote product innovation, and meet the growing and diversified demand for the tourism market (Xing, 2011) [1]. At the same time, tourism investment plays an important role in narrowing the spatial differences in the tourism economy and achieving a balanced development of regional tourism (Li et al., 2016) [2]. The scale, density, and spatial strategy of tourism investment are vital to guide the flow of tourism elements and affect the spatial pattern of regional tourism development (Li et al., 2018) [3]. Therefore, it is of great significance to explore the spatial and structural characteristics of tourism investment to optimize the spatial structure of tourism investment and realize sustainable development.

At present, China has the world’s largest domestic and outbound tourism market, as well as the fourth-largest inbound tourism market. As the scale of China’s tourism investment continues to expand under the impetus of strong tourism market demand, advantageous tourism industry policies, and diversified investment subjects, the cultural and tourism industry will become a hot and promising spot for social investment (Dai, 2018) [4]. According to the World Tourism and Travel Council (WTTC), China’s total amount of tourism investment in 2018 ranked second in the world, as it has attracted diversified capital from government investment and financing platforms, private enterprises, non-tourism enterprises, etc. It reached USD 161.5 billion, up 4.40% compared to the same period. However, some issues also emerged in tourism investment, such as a mismatch between supply and demand, investment misalignment, space aggregation, and differences in investment affect space, industry, and regional uneven development (Wang, 2021) [5]. The global financial crisis (2018) and COVID-19 (2020) have had a significant impact on economic development in China, especially tourism development. China has implemented the strategies of expanding domestic demand to promote economic growth, constructing a new development pattern with the domestic cycles as the main body and the domestic and international double cycles promoting each other. In order to give play to the important role of tourism in expanding domestic demand, the tourism investment promotion plan is implemented to stimulate the tourism consumption potential (Zeng and Cheng, 2021) [6].

The existing research results show that scholars explore the spatial-temporal pattern and evolutionary process of tourism investment by using geo-statistical methods (Yan and Chen, 2017; Su et al., 2017a; Liu, 2017) [7,8,9]. Research on the spatial structure of tourism destinations mainly focuses on the spatial network of tourist attractions and the tourism economic network of cities and provinces by using the social network method (Lee et al., 2018; Wang et al., 2020; Wang and Xia, 2018) [10,11,12], while research using the social network method to study the spatial correlation network of tourism investment is less common. From the perspective of social networks, the paper reveals the evolution process of tourism investment spatial correlation networks by analyzing the structural characteristics of tourism investment spatial correlation networks in China in four time sections and attempts to enrich the research content of tourism investment from the perspective of tourism spatial networks.

The essay has been organized in the following way. The first part introduces the research background. The second part is a literature review on the spatial structure of tourism investment and the social network method used in the tourism space structure. The third part describes the research method and data source, introducing the modified gravity model and the social network method. The fourth part reviews the empirical research, including the overall and ego network and the core-periphery model analysis. The fifth part includes the conclusion and implications.

## 2. Literature Review

### 2.1. Tourism Investment Spatial-Temporal Pattern

The evolutionary law of spatial-temporal tourism investment is one of the key points of tourism investment research (Wu, 2018) [13]. Scholars have analyzed the development trend, spatial difference, and spatial pattern of tourism fixed asset investment from the time and space perspective, taking the tourism industry or hotels and scenic spots as research objects. Endo (2006) showed a significant increase in FDI in tourism represented by hotels and restaurants; the vast majority of FDI flowed to developed countries, while it is more critical in tourism in developing countries [14]. Scholars have analyzed the spatial-temporal pattern of tourism investment in China by using geographical and economical methods. Research shows that the scale of tourism investment in China has increased rapidly and that there are significant regional differences, showing a spatial pattern of high east-west and low central tourism investment. The level of tourism investment shows a strong negative correlation between space and decreasing spatial concentration (e.g., Wu, 2001; Li, 2012; Yan and Chen, 2017; Su et al., 2017a; Su and Zhu, 2019) [7,8,15,16,17].

Some scholars took hotels as the research objects and found that China’s hotel investment was relatively concentrated in the eastern regions, less distributed in the central and western regions, and structural surplus in some regions, leading to the poor efficiency of hotel industry investment (Chen, 2006) [18]. Liu (2017) found that Star hotel investment spread from the eastern coastal areas to the central area and from the Yangtze River Delta to the middle and lower reaches of the Yangtze River [9]. Zhang (2015) took scenic spots as the research objects and pointed out that the distribution of investment in tourist attractions in China tended to be balanced, was relatively concentrated in the eastern region, had a continuous rise of humanistic and scenic spot investment in the central region, and the investment costs were higher in the western region due to the limitation of infrastructure conditions [19]. Under the influence of COVID-19, the development of cultural tourism was focused around the metropolitan area and destinations, and the consumer market sunk to the surrounding second-tier and third-tier cities (Wang, 2021) [5]. Lisiak-Zieli’nska (2021) built a tourism investment potential evaluation system and used a geo-statistical method to assess spatial changes in tourism and tourism investment potential in Stazovsky County [20].

### 2.2. Application of Social Network Analysis in the Spatial Structure of Tourist Destinations

Social network analysis is regarded as an excellent paradigm for tourism space research (Scott et al., 2008) [21]. Scholars obtain tourist flow data through questionnaires and interviews, build tourism destination spatial networks, such as scenic spots and cities, and analyze the structural characteristics, evolutionary processes, influencing factors, and marketing models of destination spatial networks.

Pavlovich (2003) explored the evolution and transformation of the tourist destination network by taking Waitomo Caves in New Zealand as an example, and discovered that the higher the density, the stronger the cohesiveness of the destination [22]. Stienmetz and Fesenmaier (2015) built a network of attractions in Baltimore to estimate the value of attractions and their connection paths [23]. D’Agata et al. (2013) and Asero et al. (2016) built a spatial network of tourist destinations in Sicily and found the mobility of tourists affected the shape, dimensions, and structure of the network [24,25]. Garcia-Palomares et al. (2015) used the social network method to assess the spatial network characteristics of attractions in eight popular European tourist cities by photo sharing and GIS spatial statistics [26]. Kang et al. (2018) and Lee et al. (2018) built a spatial network of attractions in Seoul based on social network technologies of spatial statistics and evaluated the centrality and spatial structure of attractions [10,27].

Hwang et al. (2006) analyzed the multi-cities travel pattern and network structure of international tourists in the United States, explored the role of cities among tourists in different regions, and proposed to establish a business model of tourist destination bundling [28]. Bhat and Milne (2008) conducted a survey of the embedding, density, and centrality of destination marketing networks in New Zealand, and found that the centrality of players and the structural characteristics of the overall network affected cooperative marketing [29]. Scott et al. (2008) used centrality and other standards as criteria to compare the network structure and network cohesion of four different types and development stages of tourist destinations in Australia, pointing out that the higher degree of industrialization and the larger scale, the stronger cohesion of destinations [30].

Bianchi et al. (2020) adopted the social network analysis (SNA) to study the policy network, focused on the coproduction process of tourist public policies for the disabled by looking at the network that facilitates communication among the actors taking part in the process [31]. Seok et al. (2021) explored how the structure of international tourism has changed longitudinally by using a social network analysis [32].

At the same time, scholars build the tourism destination relationship matrix by using the modified gravity model and study the characteristics, evolution processes, and network effects of tourist destination spatial networks by using the social network methods from different spatial scales, taking villages, counties, cities, and provinces as research units. Lee et al. (2013) and Ju and Tao (2016) analyzed the centrality and structure of villages using a modified gravity model, social network, and GIS technology [33,34]. Liu et al. (2021) analyzed the characteristics and influencing factors of the tourism economy spatial network by taking counties as research units [35]. Fang et al. (2014), Yu et al. (2015), and Wang et al. (2020) analyzed the spatial structure characteristics, evolutionary process, and role positioning of urban tourism economic networks and explored the spatial development model [11,36,37]. Yu et al. (2014), Wang et al. (2015), Shi et al. (2018), and Wang et al. (2019) analyzed the characteristics, evolutionary process, and spatial development model of tourism economy spatial networks in metropolitan areas and urban agglomerations [38,39,40,41]. In addition, Ma and Long (2017) and Wang and Xia (2018) took provinces as research units to analyze the characteristics of China’s interprovincial tourism economic spatial network structure and its influencing factors [12,42].

From the perspective of geography, scholars measure the spatial differences in tourism investment using geographical concentration, standard deviation, coefficient of variation, primacy index, Theil coefficient, and entropy index to analyze the spatial-temporal pattern of tourism investment. However, the social network method is seldom used to analyze the spatial connection network of tourism investment. From previous studies, scholars have used the social network method to analyze the structure, evolution, and effect of tourism destination networks, but there are few research results on tourism investment spatial correlation networks. Through the research of this paper, need to analyze the following questions: (1) What is the structure of the tourism investment spatial correlation network in China? What are the characteristics of the overall network and what is the evolution trend? (2) What is the centrality of each node and the change trend? (3) What are the nodes in the core area and the edge area, and analyze the reasons.

## 3. Materials and Methods

### 3.1. Study Area and Data Source

The samples were collected in 31 provinces (autonomous regions and municipalities directly under the Central Government) in mainland China, excluding Hong Kong, Macao, and Taiwan. The number of domestic tourists was 4.44 billion, with a revenue of RMB 3.94 trillion in China In 2016, up 11% and 15.2%, respectively, over the previous year. The number of inbound tourists reached 138 million, and the international tourism revenue reached USD 120 billion, up 3.5% and 5.6%, respectively, over the previous year. The total revenue of tourism was RMB 4.69 trillion in 2016, up 13.6% year on year. The comprehensive contribution of the national tourism industry to GDP throughout the year was RMB 8.19 trillion, accounting for 11.01% of the total GDP. Moreover, 28.13 million people were directly employed in tourism, and 79.62 million people were directly and indirectly employed in tourism, accounting for 10.26% of the total employed population in China.

Tourism investment takes many forms, of which fixed asset investment and human resources investment are the core. Based on the existing tourism investment statistics, the paper adopts the concept of tourism investment in a narrow sense (Su and Zhu, 2019) [17], that is, the fixed assets of the tourism industry in each year are used as the proxy variable of the scale of tourism investment, excluding the management and operating expenses of the maintenance and development of the industry that is transformed from part of tourism investment, as well as other tourism accommodation investments, such as private hotels, and the investment in tourism by enterprises in other industries. This paper uses the original value of tourism fixed assets as the observation index of the tourism investment scale. Some scholars use the number of students in the tourism management major and the investment in education funds to measure human capital in the tourism industry (Wang, 2015; Sheng and Liu, 2020) [43,44]. Because the employment rate of tourism management graduates in the tourism industry is low, this paper attempts to use the employees of tourism enterprises (referring to the number of employees paid by enterprises at the end of the year) to characterize human resource investment.

The tourism industry includes more than 20 industries, including accommodation, catering, transportation, sightseeing, and entertainment. Among them, hotels, tourist attractions, and travel agencies are the three core sectors of the tourism industry, which gather funds, human resources, technology, and other resources for tourism investment. This has advantages both in the absolute amount and in the attractiveness and potential of tourism for investment, showing strong typical representativeness and reflecting regional tourism industry investment (Li, 2019) [45].

Considering the consistency and accessibility of statistical calibers, the temporal span of the sample data was from 2000 to 2016, and the investment data of tourism fixed assets and human resources in the three major industries of travel agencies, tourism hotels, and tourist attractions were sorted out. Among them, travel agencies refer to the travel agencies included in the tourism statistics system, tourist hotels refer to hotels with star ratings, and tourist attractions include A-level scenic spots and non-A-level scenic spots. The original value of fixed assets and employees of travel agencies, tourist hotels, and tourist attractions in 31 provinces in mainland China was derived from the Tourism Statistics Yearbook in China (2001–2017) (copy). Among them, although the 2010 tourist attraction data were missing, the interpolation method was used to calculate and replace the missing value according to the average annual growth rate of each province and city. Due to fewer missing values, the overall analysis results were less affected.

### 3.2. Methodology

#### 3.2.1. Determination of the Tourism Investment Spatial Correlation Relationship

The determination of relationships is the key to network analysis (Scott, 2013) [46] and mainly includes three methods according to the literature. The first method was to build a relationship matrix based on the actual flow data of trade flow, investment flow, and tourism flow and analyze network structure characteristics and influencing factors (Zhong and Qin, 2017; Li et al., 2020; Peng et al., 2014) [47,48,49]. The second method was to conduct a Granger causality test according to the time series data or VAR model of regional economic and social development to determine whether there was a correlation between two regions, build a regional directed spatial correlation network, and analyze the characteristics of the spatial network structure and driving factors (Liu et al., 2015; Li et al., 2014; Huang, 2018) [50,51,52]. The third method was to use the gravitation model to determine the connection strength between two regions, which is widely used in the urban economy, tourism economy, population mobility, energy consumption, and other fields, such as urban agglomeration spatial economic network (Wang and Yang, 2018) [53], metropolitan tourism economic network (Yang et al., 2018) [54], population migration network (Wang et al., 2014) [55], energy consumption spatial connection network (Liu et al., 2015) [56], and other research. The paper uses the gravity model method to determine the spatial correlation of tourism investment, mainly because the gravity model is not only applicable to the total data, but it can also take into account the spatial distance factor and then use cross-sectional data to depict the evolution trend of the spatial correlation network.

In the research field of tourism economic spatial structure, scholars use the modified gravity model to measure the strength of tourism economic linkages and analyze the spatial structure characteristics of tourism economic linkage networks. Usually, a tourism economic connection gravity model is built based on the number of tourists, tourism income, and the spatial distance between cities to measure the gravity intensity between tourism cities and build a symmetric gravity matrix (Zou, 2017; Yang et al., 2018) [54,57]. Some scholars use tourism income or tourism resources to build a gravitational coefficient into the gravitational model and build an asymmetric gravitational matrix (Wang et al., 2017; Yan et al., 2018) [58,59] or use the per capita GPD to modify the spatial distance to build an economic distance indicator to measure the tourism economic contact strength (Wang and Xia, 2018) [12]. In the process of tourism investment research, scholars mainly focus on tourism fixed asset investment and human capital investment (Su and Zhu, 2019; Sheng and Liu, 2020) [17,44]. In combination with the characteristics of tourism investment spatial correlation, the gravity model of tourism economic linkages was revised (Ma and Long, 2017; Zou et al., 2018) [42,60], using two indicators of tourism enterprises’ human resource investment and fixed asset investment and using tourism fixed asset investment to build the relationship coefficient *k_ij_* (Dong et al., 2018) [61], as shown in Formula (1). The modified gravity model of tourism investment spatial correlation was used to measure the spatial correlation degree of interprovincial tourism investment.
(1)$$F_{ij}=k_{ij}\frac{\sqrt{H_{i}*I_{i}}*\sqrt{H_{j}*I_{j}}}{D_{ij}^{2}},\;(k_{ij}=\frac{I_{i}}{I_{i}+I_{j}})$$

In the formula, *F_ij_* is the tourism investment spatial correlation degree of province *i* to province *j*; *H_i_*, *H_j_* and *I_i_*, *I_j_* are the human resources investment and fixed asset investment in tourism enterprises in province *i* and *j*, respectively. Fixed asset investment uses the original value of fixed assets (RMB 10,000) of the three core departments, including travel agencies, hotels, and scenic spots. The investment of human resources is measured by employees (10,000 people) in the three core sectors of the tourism industry. *D_ij_* is the spherical distance (km) between provincial capital cities *i* and *j*, using the spatial friction index 2 commonly used by scholars. Based on the calculation relationship coefficient *k_ij_* of tourism fixed assets investment, an asymmetric matrix of tourism investment spatial correlation was constructed. On the basis of measuring the spatial correlation degree of tourism investment, the spatial correlation degree of tourism investment was divided up between a province and the others to obtain the tourism investment spatial correlation amount of province *C_i_* (Zou et al., 2018) [60]:(2)$$C_{i}=\overset{n}{\underset{j=1}{\sum }}F_{ij}$$

The gravitational matrix interprovincial tourism investments was calculated according to Formula (1), and the average value after eliminating the gravitational outliers was recorded as the segmentation value of the tourism investment spatial correlation network (Liu et al., 2010; Wang et al., 2014) [55,62]. If the gravitational value was greater than or equal to the segmentation value, it was recorded as 1, indicating that the province in this row had an association/relationship with the tourism investment in this list of provinces. If the gravitational value was less than the segmentation value, it was recorded as 0, indicating that there was no correlation relationship. Using UCINET 6.0 to binarize the gravity matrix, a directional network of tourism investment spatial correlation was obtained.

#### 3.2.2. Social Network Method

The social network method is one of the more practical ways of providing a quantitative analysis of various relationships and revealing the structure of relationships between research objects. Overall network characteristics are depicted by network density, network correlation, network hierarchy, and network efficiency (Liu, 2019) [63]. Network density represents the tightness of spatial correlation between network nodes and is calculated by dividing the number of actual relationships in the network by the theoretical maximum number of relationships; the value range is [0, 1]. The greater the network density is, the closer the spatial correlation, the greater the impact of the network structure on the actors, and the more stable the network. The network correlation degree reflects the robustness of the network. For a directed graph, if any two points can be connected, it is called the correlation graph. The network correlation degree is measured by “accessibility”. Network hierarchy measures the extent to which nodes in the network can reach each other asymmetrically, reflecting the hierarchical structure of nodes in the network. The higher the degree of network hierarchy, the stricter the hierarchical structure between nodes, and more nodes are in subordinate and marginal positions in the network. Network efficiency reflects the connection efficiency between nodes. The more connections there are between nodes, the lower the network efficiency and the more stable the network.

Ego network characteristics are measured by centrality to describe the position of nodes in the network, including degree centrality, betweenness centrality, and closeness centrality (Jiang et al., 2019) [64]. Degree centrality measures the central position of a node in the network, which was measured based on the number of nodes directly connected to the node, neglecting to consider the indirectly connected nodes. The more direct linkages a node has with other members, the more central position it holds in the network, and the more power it has. To compare the size of the degree centrality of points in different graphs, Freeman (1979) proposed relative degree centrality as the ratio of the absolute centrality of a node to the maximum possible degree of a vertex. In a directed network, the calculation formula of the relative degree centrality of node *i* is as follows:(3)$$C_{RD}\left(i\right)=\frac{C_{DI}\left(i\right)+C_{DO}\left(i\right)}{2n-2}$$

In the formula, *C_RD_(i)* is the relative degree centrality of node *i*. *C_DI_(i)* is the in-degree of node *i*, which is the number of other nodes entering node *i*, that is, the number of direct relationships obtained by this node. *C_DO_(i)* is the out-degree of node *i*, which is the number of relationships directly spread by this node. n is the number of nodes.

Betweenness centrality is used to evaluate the control degree over various resources by the nodes in the network. As the shortest intermediary between more nodes, the higher the intermediate centrality of a node is, the more structural holes it has, and the stronger the control ability of nodes over network resources.
(4)$$C_{RB}\left(i\right)=\frac{2C_{AB}\left(i\right)}{\left(n-1\right)\left(n-2\right)}=\frac{2\sum_{j}^{n}\sum_{k}^{n}g_{jk}\left(i\right)/g_{jk}}{n^{2}-3n+2}\;(j\neq k\neq i,\;and\;j<k)$$

In this formula, *C_AB_*(*i*) and *C_RB_*(*i*) are the absolute and relative betweenness centrality of node *i*. *g_jk_* is the number of shortest linking paths between *j* and *k*, while *g_jk_*(*i*) represents the number of shortest linking paths between *j* and *k* passing through node *i*.

Closeness centrality reflects the closeness degree of the nodes linked to each other and is measured by the shortest distance between a node and others. The larger the relative closeness centrality of a node, the closer the node links with each other. The node is not controlled by the others and is in the core position in the network.
(5)$$C_{RC}\left(i\right)=\frac{n-1}{C_{AC}^{-1}\left(i\right)}=\frac{n-1}{\sum_{j=1}^{n}d_{ij}}$$

In this formula, $C_{AC}^{-1}\left(i\right)$ and $C_{RC}\left(i\right)$ are the absolute and relative closeness centrality of node *i*, respectively, and *d_ij_* is the shortcut distance between nodes *i* and *j* (the number of lines is included in the shortcut).

The core-periphery model shows which nodes are in the core or peripheral areas and reveals the internal relationship between the nodes in the core-periphery regions. The dynamic change process of nodes both in the core peripheral areas is analyzed through cross-section comparison. UCINET 6.0 was used to conduct a module analysis on the tourism investment spatial correlation network.

## 4. Results and Discussion

### 4.1. Number of Tourism Investment Spatial Correlations

Based on the revised gravity model, the spatial correlation degree of interprovince tourism investment was measured to obtain the spatial correlation number of interprovinces (shown in Table 1). 

The survey results show that (1) the overall spatial correlation number of interprovince tourism investment shows a trend of continuous growth at an increase of 242.95% from 5440.20 to 18,656.97. Driven by the development strategies of the Eastern region in taking the lead in development, Western Development, the rise of the Central region, and revitalization of the Northeast, the scale and quality of tourism investment continue to improve. The spatial correlation degree of interprovince tourism investment shows an increase, but there are spatial differences. (2) The provinces with higher and lower tourism investment spatial correlations are relatively stable. The top-ranked provinces, including Beijing, Jiangsu, Zhejiang, Shanghai, Shandong, and Guangdong, are concentrated in the economically developed regions in East China, while the lower-ranked provinces, including Ningxia, Qinghai, Xinjiang, Xizang, Heilongjiang, and Guizhou, are concentrated in West and Northeast China, which are far from the main source markets in the eastern regions constrained by economic levels and transportation accessibility. (3) Although the unbalanced pattern of regional tourism investment has not changed, the trend of development is increasing. The proportion of the total tourism investment spatial correlation in the top five provinces fell from 75.53% to 64.36%, while the proportion of the bottom five provinces rose from 0.23% to 0.41%. The gap in inter-province tourism investment spatial correlation shows a decreasing trend, mainly due to regional economic development strategies and social capital entering the field of tourism investment.

### 4.2. Overall Network Structure Characteristics

According to the modified gravity model, the spatial correlation of provincial tourism investment was determined, and a relationship matrix was established. The UCINET 6.0 visualization tool was used to draw the spatial correlation network of tourism investment in 2000, 2005, 2010, and 2016. Figure 1 shows that the center of China’s tourism investment spatial correlation network is Beijing, Shandong, Guangdong, Jiangsu, Zhejiang, and Shanghai. Isolated points included Qing hai, Xizang, Ningxia, and Xinjiang in 2000 and only included Xizang and Xinjiang in 2016. Overall, the structure of China’s tourism investment spatial correlation network is becoming increasingly complex, and the tourism investment spatial correlation is becoming increasingly closely related.

The higher the network density is, the stronger the linkages between nodes, the more ways to obtain linkages from other nodes, and the more conducive it is to the development of each node. The overall network density and network association of China’s tourism investment spatial correlation showed an increasing trend (shown in Figure 2), network associations increased from 195 to 258, and the network density increased from 0.210 to 0.277, indicating increasingly close linkages between provinces.

The data show that 2009 can be divided into two phases as a demarcation point. The first stage was the period of steady growth from 2000 to 2009, with the network density and network correlation increasing from 0.210 and 195 to 0.302 and 281, respectively, which indicates that tourism investment is increasingly closely linked with the growth of the tourism investment scale. Moreover, 2010–2016 is a period of volatile development. The U.S. subprime mortgage crisis in 2008 triggered the global economic crisis, which had some impact on the scale of China’s tourism investment, causing the density and correlation of tourism investment networks to decline in 2010 and reached a trough in 2011. To promote the development of tourism, China issued policies such as “Implementation Opinions on Encouraging and Guiding Private Capital investment in Tourism” and “Several Opinions on Financial Support for the Accelerated Development of Tourism” to adopt tourism as an important industry to alleviate overcapacity, to stimulate domestic demand and to transform industry. All the policies brought major development opportunities in the field of neotourism investment, such as online tourism, rural tourism, vacation tourism, sports tourism, and cruise tourism. In 2016, the density and correlation of tourism investment networks rose to 0.277 and 258, respectively.

The network correlation of tourism investment is measured by the connectedness, graph hierarchy, and graph efficiency. Figure 3 shows that the connectedness showed an upward trend rising from 0.755 to 0.873, which indicates that China’s tourism investment was gradually becoming closely linked and that there were obvious spatial correlations and spillover effects. The graph hierarchy showed a downward trend from 0.323 volatility to 0.133, which caused the hierarchical relationship of the spatial correlation structure to weaken and the mutual influence of tourism investment to gradually increase. The graph efficiency showed a slow downward trend of volatility from 0.655 to 0.646, indicating that the tourism investment spatial correlation network was in the increase of connectivity, and the network stability was enhanced. Comparing the three results, it can be seen that the hierarchy system of tourism investment spatial correlation is weakened as regional tourism integration progresses, the spatial correlation relationship is increased, and the stability of the network is enhanced.

### 4.3. Ego Network Characteristics

UCINET 6.0 was applied to test the degree centrality, betweenness centrality, and closeness centrality in the four time series of 31 provinces to further analyze the ego characteristics of each province in the tourism investment spatial correlation network. The calculation results and rankings are shown in Table 2.

#### 4.3.1. Degree Centrality

The degree centrality of each province has a significant growth trend, with the number exceeding the average increasing from 14 to 17 and the number of isolated nodes decreasing from 4 to 2 (Table 2, Figure 1). This illustrates that the spatial correlation of interprovincial tourism investment gradually strengthened. Although the ranking changes, the degree centrality of Beijing, Shandong, Guangdong, Jiangsu, Zhejiang, and Shanghai shows a relatively significant increase, which makes the 6 provinces firmly at the top. They have a more prominent central position in the tourism investment spatial correlation network, have the most direct linkages with other provinces, and have greater resources and power. At the same time, Hubei, Hunan, Henan, and Hebei rank in the second tier and struggle into the top ten as their increase in degree centrality is large, and the central position is further enhanced by strengthening tourism investment spatial correlation with other provinces. Qinghai, Ningxia, Xizang, and Xinjiang rank lower and are subordinate in the network, as they are far away from the eastern source market, causing weak tourism demand and a small scale of tourism investment, so that the tourism investment spatial correlation with other provinces is weak.

In Figure 4, the estimation results of the relative degree centrality kernel density of the China tourism investment spatial correlation network show that the kernel density estimation chart of the relative degree centrality was lower to the right during the two periods in 2000 and 2005, indicating that the number of provinces with more tourism investment spatial correlation is relatively small. The peak value tends to decline with the right tail lengthening and raising, and the overall distribution shifts to the right, indicating that the number of provinces with more tourism investment spatial correlation is gradually increasing and the tourism investment spatial correlation is much closer. From 2010 to 2016, the provinces with a high number of tourism investment spatial correlations increased on a large scale in the twin peaks model; meanwhile, there was also an increase in the number of provinces with a low number of tourism investment spatial correlations.

Turning now to the analysis of degree centrality potential, it depicts the overall level of investment spatial correlation of the network, reflecting the non-asymmetry and imbalance of tourism investment spatial correlation between provinces. Table 2 shows that the degree centrality potential has a downward trend from 36.38% to 30.92%. The concentration of the tourism investment network in China is decreasing, and tourism investment shows a trend of equilibrium. With the implementation of the coordinated development strategies of some regional economies, the level of tourism investment in provinces has continuously improved with the obvious reflow effect, especially in the central and western and northeast regions (Su and Zhu, 2019) [17]. The rapid growth of the tourism demand market in provinces, the increasing demand for tourism products, and the need to increase tourism investment push tourism to be one of the important channels to expand domestic demand.

#### 4.3.2. Betweenness Centrality

The results obtained from the preliminary analysis show an increase in the betweenness centrality of most provinces. The betweenness centrality of the top six provinces, including Beijing, Guangdong, Shandong, Liaoning, Shaanxi, and Hunan, was above the mean in 2016. The betweenness centrality of Shaanxi, Gansu, and Jilin has increased dramatically by a breakthrough from 0, exceeding the average and entering the top ten in 2016. It shows that the abovementioned provinces are closely related to others in space, have a higher degree of intermediary, and have a strong control and dominant role over others. The betweenness centrality of Guizhou, Tianjin, and Henan has achieved a breakthrough of 0 and showed an upward trend, indicating that nodes on the network resources of the intermediary capacity are increasing. The betweenness centrality of Nei Mongol, Heilongjiang, Hainan, Xizang, Qinghai, Ningxia, and Xinjiang was 0, indicating that the 7 provinces were not the intermediary of any other provinces, with no ability to control resources. Because of the weak economic base, small tourism market demand and tourism investment scale, and isolated geographical location away from the source market in eastern China, it is difficult for these provinces to control and dominate others in the network.

#### 4.3.3. Closeness Centrality

The results also show a substantial increase in closeness centrality, indicating strengthened tourism investment spatial correlation between provinces and increased regional influence. With a slight change in the ranking, Beijing, Guangdong, Shandong, Jiangsu, Zhejiang, and Shanghai were ranked in the forefront of the country, indicating that they had close spatial correlation with other provinces and better accessibility, were less controlled by other provinces, were the center position of the tourism investment network, and had greater influence in the region. Ningxia, Qinghai, Hainan and Xizang, Xinjiang ranked bottom, the closeness centrality of Ningxia and Qinghai was nearly 0, and that of Xizang and Xinjiang was close to the center of 0, showing that the five provinces were on the edge of the network.

### 4.4. Core-Periphery Model Analysis

Based on the binary matrix of China’s tourism investment spatial correlation network, core-periphery analysis was carried out by using UCINET 6.0 to divide 31 provinces into core-periphery areas and to measure the density of tourism investment spatial correlation in the core-periphery areas. The core areas of China’s tourism investment network demonstrate an expanding trend increasing from 10 provinces to 12 and decreasing from 21 provinces to 19 in the peripheral regions from 2000 to 2016 (Figure 5).

Spatially, the core region mainly includes provinces in the eastern and central regions, of which 7 are in the eastern regions, 4 are in the central regions, and only 1 is in the western regions. The peripheral area mainly includes provinces in West and Northeast China. Early tourism prosperity, the highest level of economic development, excellent location conditions, the super taste of tourism resources, and large spatial density make the eastern region the largest tourism market, the most active tourism investment region, and the region with the most prominent tourism economic effects in China. In particular, the Yangtze River Delta region, the Pearl River Delta region, the Beijing-Tianjin-Hebei region, the Ring Bohai region, the West Coast region, and other regional tourism integration areas have acted as the tourism investment “magnetic field” to become tourism investment aggregations. The three core sectors (travel agencies, scenic spots, and hotels) in the eastern regions account for 65% of the country’s fixed asset investment and 60% of its employees (Li et al., 2019) [45]. Although the level of economic development in the central regions is slightly lower than that in the eastern part, it has abundant tourism resources and a strong demand for the tourism market. Under the rise of central China’s strategy, there is an obvious trend of concentration of tourism investment hot spots moving toward the central regions (Yan and Chen, 2017) [7].

Western development has led to regional economic development and further improvement of tourism facilities and investment environments, becoming regions of concern to investors, and the depression effect of tourism investment in the western region has begun to appear (Su and Sun, 2017b) [65]. However, the economic status of the western region in China has not substantially changed. The scale of tourism investment is severely restricted by a weak economic base, coupled with the remoteness of the eastern source market, the slow development of the internal source market, the ecologically fragile environment, and other factors in the western regions. The proportion of tourism investment in most provinces and cities ranks at the bottom in China, such as Qinghai, Xinjiang, and Ningxia.

Although Northeast China’s industrial tourism, red tourism, ice and snow tourism, and other tourism resources are relatively rich, leading to potential tourism development, there is still a certain gap in the level of tourism development compared with other regions. Since implementing the Reform and Opening-up strategy, the economic development of Northeast China has gradually lagged behind that of the Eastern coastal regions, and the strategy of revitalization has promoted the transformation of the industrial structure of the old industrial base in the northeast, which is in the alternating stage of economic development momentum, and the scale of tourism investment has been restricted by the level of economic development.

Judging from the changing trend of the density of China’s tourism investment core-periphery structure network, the network density of core and periphery areas shows different degrees of growth, which promotes the development of the tourism investment spatial correlation network. The network density of the core areas shows fluctuating development, increasing from 0.811 in 2000 to 0.902 in 2010 and then decreasing to 0.871 in 2016. Despite the fluctuations, the tourism investment of the provinces in the core areas is closely linked. Meanwhile, although the network density of periphery areas increased from 0.057 (in 2000) to 0.091 (in 2016), the network of tourism investment spatial correlation is still not close. The linkage density between the core areas and the periphery areas decreased from 0.381 to 0.329, indicating that the tourism investment linkage between the core areas and the periphery areas was weakening.

## 5. Conclusions

The most obvious and important findings to emerge from the above analysis are as follows: (1) The spatial correlation number of interprovincial tourism investment continues to grow, with Beijing, Jiangsu, Zhejiang, Shanghai, Shandong, and Guangdong at the top of the list. (2) Overall network density and correlation are rising, and the spatial correlation of interprovincial tourism investment is becoming increasingly closely linked. Network hierarchy and network efficiency are decreasing, and network stability is enhanced. (3) The ego network structure characteristics indicate that the degree centrality and closeness centrality of each province have shown a significant increase; Beijing, Shandong, Guangdong, Jiangsu, Zhejiang, and Shanghai are the top six and in the center of the network, while Qinghai, Ningxia, Xizang, Xinjiang, and Hainan are relatively low and in subordinate locations. Most provinces have improved betweenness centrality; Beijing, Guangdong, Shandong, Liaoning, Shaanxi, and Hunan have a strong betweenness centrality with strong intermediary capacity, while the betweenness centrality of Nei Mongol, Heilongjiang, Hainan, and the other seven provinces is 0. (4) The core area mainly includes eastern and central provinces, and the periphery areas mainly include western and northeastern provinces. The network connection density of the core and periphery areas shows an increasing trend, while the network linkage density between the core and periphery areas shows a downward trend.

### 5.1. Implications

These findings have important implications for promoting the sustainable development of China’s tourism investment: (1) The result of the social network analysis of the tourism investment spatial correlation network in China can optimize the network structure of tourism investment and improve the level of regional tourism integration. The attribute data and relationship data of China’s tourism investment are basically consistent. In 2016, Guangdong, Beijing, Shanghai, Zhejiang, Jiangsu, and Shandong were the top provinces in terms of fixed asset investment and human resource investment in China’s tourism industry. According to the social network analysis, the provinces with the highest degree of centrality and closeness centrality include Beijing, Shandong, Guangdong, Jiangsu, Zhejiang, and Shanghai. The top six provinces in betweenness centrality include Beijing, Guangdong, Shandong, Liaoning, Shaanxi, and Hunan. In the process of regional tourism integration, we should pay attention not only to attribute data but also to the relationship data of the tourism investment spatial correlation network. It is a requisite to give play to the central and intermediary role of provinces, including Beijing, Guangdong, Shandong, Jiangsu, Zhejiang, Shanghai and Liaoning, Shaanxi, and Hunan, to optimize the network structure of tourism investment spatial correlation and to realize the mutual promotion of regional tourism integration and network structure. (2) The result of core-periphery analysis of the tourism investment spatial correlation network in China can lessen the tourism investment gap in core-periphery regions by bringing the functions of different provinces into play. Provinces in the core area should give full play to the advantages of capital, talent, and enterprise management, strengthen their core position in the spatial connection network of tourism investment, supply high-quality tourism products, and meet the tourism demand inside and outside the region. The marginal provinces should, according to the stage of economic development, consider ecological environment protection and tourism market demand, adjust measures to local conditions, scientifically plan and appropriately invest in tourism to increase the supply of tourism products to promote the high-quality development of the tourism industry. The density of tourism investment spatial correlation between core-periphery areas, which is affected by the level of economic development, population, tourism resources, transportation accessibility, and urbanization, shows a downward trend. We advise them to exert the spatial spillover effect of tourism investment in the core areas, to pull the tourism investment of the periphery areas, to realize the sharing of tourism resources and the tourism market, and to promote common development.

As spatial analysis has been a great ally of tourism planning for most countries, the findings also highlight the need for a strategy that allows for long-term sustainable tourism development.

### 5.2. Limitations and Future Research Directions

The paper analyzes the tourism investment spatial correlation network based on tourism investment stock data and does not use bilateral tourism investment data. In follow-up research, we try to use enterprise investment flow data to build a bilateral tourism investment flow matrix to better analyze the characteristics of the tourism investment network.

Meanwhile, the influencing factors of tourism investment involve the economic development level, market demand, resource endowment, and accessibility. Scholars generally construct econometric models to analyze the key factors affecting tourism investment. There is some room for further progress in making empirical research by the QAP method from the perspective of social networks to analyze the influencing factors of tourism investment networks.

## Figures and Tables

**Figure 1 ijerph-19-15661-f001:**
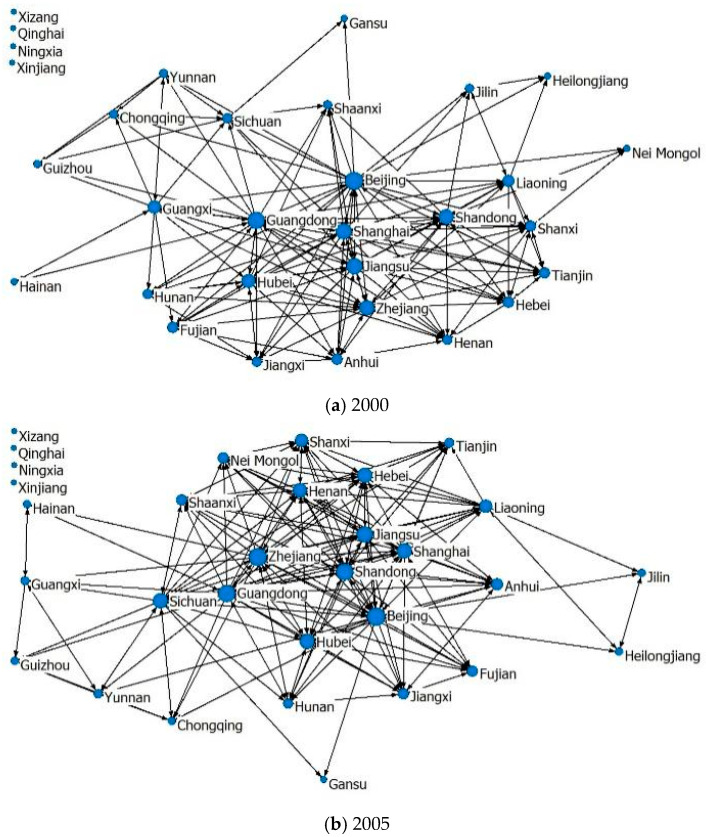
Spatial correlation network of tourism investment in China from 2000 to 2016 (**a**–**d**).

**Figure 2 ijerph-19-15661-f002:**
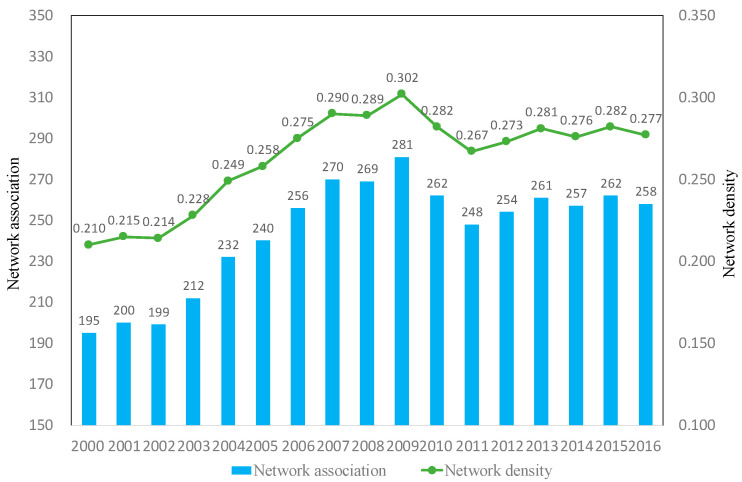
Network density and network association number.

**Figure 3 ijerph-19-15661-f003:**
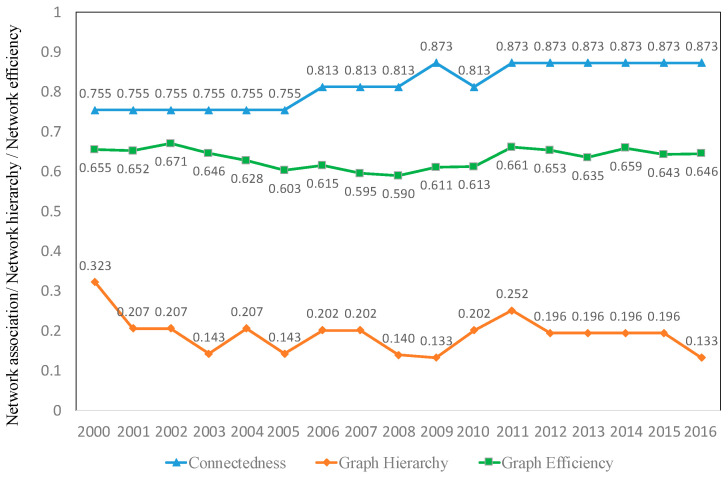
Network relevance.

**Figure 4 ijerph-19-15661-f004:**
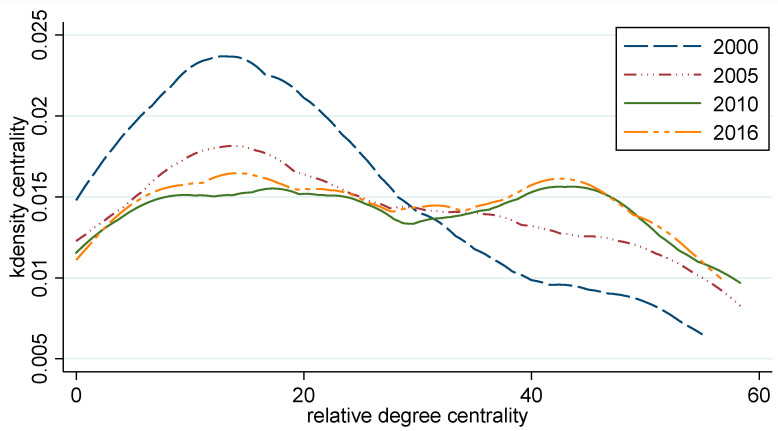
Kernel density estimation of point degree centrality.

**Figure 5 ijerph-19-15661-f005:**
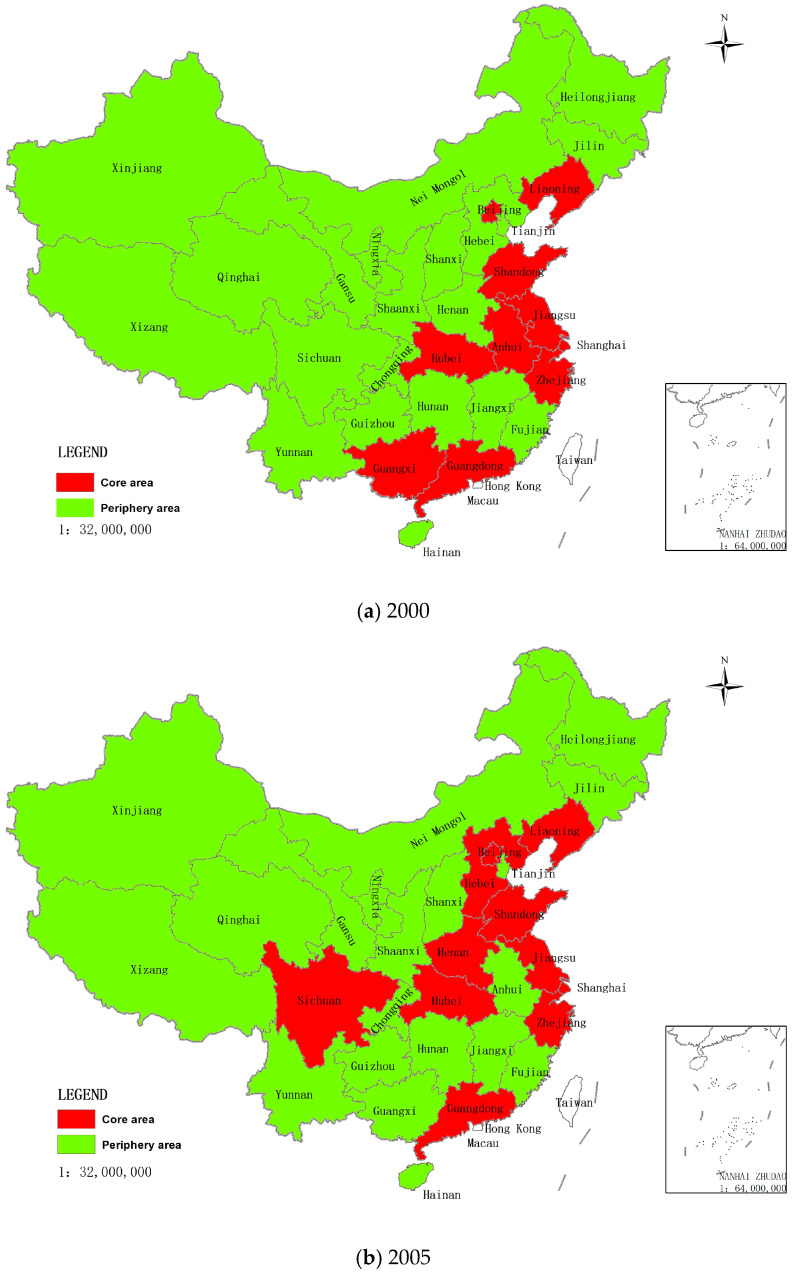
Core-periphery structure of the tourism investment spatial correlation network in China from 2000 to 2016.

**Table 1 ijerph-19-15661-t001:** Spatial correlation quantity of tourism investment in each province of China from 2000 to 2016.

Province	2000		2005		2010		2016	
	Number	Percentage (%)	Number	Percentage (%)	Number	Percentage (%)	Number	Percentage (%)
Beijing	1320.77	24.28	2758.94	19.82	2873.67	16.39	2889.89	15.49
Jiangsu	620.94	11.41	1758.84	12.63	2440.99	13.92	2851.80	15.29
Zhejiang	579.00	10.64	2052.37	14.74	2298.28	13.11	2626.17	14.08
Shanghai	1080.24	19.86	2031.41	14.59	2094.55	11.95	2397.36	12.85
Shandong	244.16	4.49	882.89	6.34	1406.72	8.02	1243.68	6.67
Guangdong	508.47	9.35	910.23	6.54	1163.18	6.63	1220.36	6.54
Hebei	50.42	0.93	615.32	4.42	692.18	3.95	659.78	3.54
Anhui	105.78	1.94	194.65	1.40	501.82	2.86	644.78	3.46
Hubei	111.87	2.06	330.96	2.38	543.41	3.10	539.99	2.89
Henan	17.93	0.33	245.99	1.77	412.61	2.35	521.50	2.80
Hunan	34.33	0.63	137.86	0.99	529.80	3.02	400.10	2.14
Sichuan	65.96	1.21	412.71	2.96	292.68	1.67	383.12	2.05
Shanxi	53.33	0.98	212.31	1.52	287.78	1.64	324.07	1.74
Fujian	49.88	0.92	97.71	0.70	162.19	0.93	264.96	1.42
Shanxi	22.73	0.42	100.33	0.72	177.15	1.01	225.79	1.21
Tianjin	94.76	1.74	130.12	0.93	142.02	0.81	207.54	1.11
Chongqing	24.45	0.45	92.95	0.67	146.15	0.83	181.78	0.97
Jiangxi	20.74	0.38	95.99	0.69	335.18	1.91	167.98	0.90
Yunnan	30.75	0.57	97.15	0.70	122.61	0.70	160.93	0.86
Liaoning	136.89	2.52	300.78	2.16	327.06	1.87	154.56	0.83
Hainan	34.14	0.63	93.62	0.67	89.49	0.51	136.18	0.73
Guangxi	133.34	2.45	111.23	0.80	122.01	0.70	92.74	0.50
Nei Mongol	6.96	0.13	69.36	0.50	89.11	0.51	90.98	0.49
Gansu	8.37	0.15	25.76	0.19	73.65	0.42	79.61	0.43
Guizhou	11.02	0.20	25.32	0.18	32.81	0.19	71.03	0.38
Jilin	51.74	0.95	64.63	0.46	82.09	0.47	43.90	0.24
Heilong-jiang	15.16	0.28	56.70	0.41	49.86	0.28	27.38	0.15
Ningxia	1.17	0.02	3.68	0.03	12.12	0.07	19.65	0.11
Qinghai	0.78	0.01	2.71	0.02	7.42	0.04	13.69	0.07
Xinjiang	3.76	0.07	8.63	0.06	22.22	0.13	13.34	0.07
Xizang	0.37	0.01	0.95	0.01	1.20	0.01	2.31	0.01
Max.	1320.77	24.28	2758.94	19.82	2873.67	16.39	2889.89	15.49
Min.	0.37	0.01	0.95	0.01	1.20	0.01	2.31	0.01
Ave	175.49	3.23	449.10	3.23	565.55	3.23	601.84	3.23
Total	5440.20	100	13,922.10	100	17,532.03	100	18,656.97	100

Note: provinces are ranked according to the 2016 calculation. Source: the result of author calculation.

**Table 2 ijerph-19-15661-t002:** Centrality of the tourism investment spatial correlation network in China from 2000 to 2016.

Province	Degree Centrality (%)				Province	Betweenness Centrality (%)				Province	Closeness Centrality (%)			
	2000	2005	2010	2016		2000	2005	2010	2016		2000	2005	2010	2016
Beijing	55.00	58.33	56.67	56.67	Beijing	10.09	10.89	8.14	15.58	Beijing	19.74	19.61	24.00	31.58
Shan- dong	43.33	51.67	58.33	56.67	Guang- dong	8.75	9.21	12.15	14.49	Guang- dong	19.48	19.48	24.00	30.93
Guang- dong	51.67	55.00	58.33	55.00	Shan- dong	4.50	3.12	6.44	8.08	Shan- dong	18.63	18.99	23.81	30.00
Jiangsu	46.67	50.00	53.33	53.33	Liao- ning	6.50	7.09	7.01	7.15	Jiangsu	18.99	18.99	23.26	30.00
Zhejiang	43.33	58.33	51.67	48.33	Shaanxi	0.00	0.13	1.67	5.21	Zhe- jiang	18.75	19.35	23.08	29.70
Shang- hai	46.67	45.00	45.00	46.67	Hunan	0.08	0.11	4.43	4.41	Shang- hai	19.11	18.99	22.56	29.70
Henan	16.67	40.00	48.33	46.67	Gansu	0.00	0.00	2.87	3.74	Henan	18.07	18.63	22.90	29.41
Hubei	36.67	43.33	48.33	46.67	Sichuan	4.00	8.48	2.53	3.62	Hunan	17.86	17.96	22.90	29.13
Hunan	16.67	21.67	48.33	46.67	Jiangsu	4.40	2.09	2.72	3.25	Hubei	18.40	18.52	22.73	29.13
Hebei	23.33	43.33	41.67	38.33	Jilin	3.04	0.00	0.08	3.10	Sichuan	18.07	18.99	22.56	29.13
Shanxi	11.67	25.00	35.00	38.33	Chong- qing	0.11	0.23	0.52	2.58	Shaanxi	17.75	18.18	22.56	28.85
Anhui	25.00	28.33	35.00	36.67	Zhe- jiang	3.70	7.93	2.95	1.97	Hebei	18.07	18.63	22.39	28.30
Sichuan	20.00	45.00	31.67	35.00	Henan	0.00	1.13	2.46	1.96	Chong- qing	17.65	17.65	21.90	28.30
Shanxi	25.00	31.67	31.67	30.00	Hebei	0.08	2.16	1.77	1.49	Shanxi	18.18	18.40	22.22	28.04
Fujian	23.33	23.33	30.00	30.00	Hubei	6.46	1.64	1.44	1.37	Anhui	18.07	18.07	22.06	28.04
Chong- qing	11.67	11.67	23.33	30.00	Guangxi	8.60	0.61	0.50	1.28	Jiangxi	17.96	18.07	21.90	27.78
jiangxi	18.33	25.00	33.33	28.33	Shang- hai	3.37	1.43	1.21	1.24	Fujian	17.96	17.86	21.74	27.52
tianjin	21.67	20.00	16.67	20.00	Yunnan	4.93	0.36	0.11	1.10	Gansu	17.14	17.14	21.28	26.79
Liaoning	28.33	35.00	33.33	18.33	Shanxi	0.63	0.49	0.64	0.73	Tianjin	17.96	17.86	20.69	26.32
Nei Mongol	5.00	23.33	15.00	15.00	Anhui	0.06	0.23	0.15	0.28	Liao- ning	18.18	18.40	22.06	26.09
Guangxi	30.00	15.00	16.67	15.00	Guizhou	0.00	0.00	0.00	0.21	Nei Mongol	17.14	18.18	20.55	25.64
Guizhou	8.33	8.33	10.00	15.00	Fujian	0.29	0.11	0.02	0.13	Guangxi	18.40	17.44	20.69	25.21
Yunnan	13.33	13.33	13.33	15.00	Tianjin	0.00	0.00	0.00	0.11	Guizhou	17.24	17.24	20.55	25.21
Gansu	3.33	3.33	10.00	11.67	Jiangxi	0.04	0.13	0.07	0.01	Yunnan	17.65	17.65	20.41	25.00
Jilin	11.67	8.33	10.00	8.33	Nei Mongol	0	0	0	0	Jilin	17.34	17.05	20.27	25.00
Heilong-jiang	6.67	8.33	8.33	6.67	Heilong-jiang	0	0	0	0	Heilong-jiang	17.14	17.05	20.13	25.00
Hainan	6.67	8.33	8.33	6.67	Hainan	0	0	0	0	Ningxia	-	-	-	25.00
Ningxia	0.00	0.00	0.00	3.33	Ningxia	0	0	0	0	Hainan	16.85	17.05	20.13	24.39
Qinghai	0.00	0.00	1.67	1.67	Qinghai	0	0	0	0	Qinghai	-	-	17.96	21.58
Xizang	0.00	0.00	0.00	0.00	Xizang	0	0	0	0	Xizang	-	-	-	-
Xinjiang	0.00	0.00	0.00	0.00	Xinjiang	0	0	0	0	Xinjiang	-	-	-	-
mean	20.96	25.8	28.17	27.74	mean	2.24	1.85	1.93	2.68	mean	18.06	18.2	21.83	27.47
centrality (%)	36.38	34.77	32.24	30.92	centrality (%)	8.11	9.33	10.56	13.33					

Note: provinces are ranked according to the 2016 calculation. Source: the result of author calculation.

## Data Availability

The data presented in this study are available on request from the corresponding author.

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
