# Peer review of "Research on the Structural Characteristics and Evolutionary Process of China’s Tourism Investment Spatial Correlation Network"

_ijerph, 2022, doi:10.3390/ijerph192315661_

Round 1

Reviewer 1 Report

Dear Authors, Thank you very much for your work. It sounds interesting, informative, and topical.

Your abstract is clear, but it runs directly to the results, not passing through the framework or the methods.

The so-called literature review is also minimalist and focused on the China reality. Some international background and information would sound appropriate.

I also call your attention to the illustrations. The tables are difficult to read because some titles are too long and the numbers are easier to read (and compare) when they are aligned to the right and not centred. The same with figures number 1 to 3, which have too much side and inside information (too 'noisy').

Notwithstanding my comments, in the end, I really appreciate your work, and I believe it will be very well received by the scientific (and maybe also non-scientific) community.

Author Response

Dear reviewer:

Thank you very much for your affirmation of the paper and your valuable suggestions for revision. According to the modification opinions, we made serious modifications, as follows:

1.Your abstract is clear, but it runs directly to the results, not passing through the framework or the methods.

In the abstract, the research methods used in the paper are supplemented.

see lines 8-11.

The paper uses the revised gravity model to measure the intensity of tourism investment spatial correlation, constructs the spatial correlation matrix of tourism investment, and uses social network method to analyze the structural characteristics and evolutionary process of tourism investment spatial correlation network based on 31 provinces in China from 2000 to 2016.

2.The so-called literature review is also minimalist and focused on the China reality. Some international background and information would sound appropriate.

Relevant literature on tourism flow research using social network method is supplemented,

see lines 113-139

  Pavlovich (2003) explored the evolution and transformation of tourist destination network by taking Waitomo Caves in New Zealand as an example, and discovered that the higher the density, the stronger the cohesiveness of the destination[22]. Stienmetz and Fesenmaier (2015) built a network of attractions in Baltimore to estimate value of attractions and their connection paths[23]. D'Agata et al. (2013) and Asero et al. (2016) built the spatial network of tourist destination in Sicily, and found the mobility of tourists affecting the shape, dimensions and structure of the network[24,25]. Garcia-Palomares et al. (2015) used social network method to assess the spatial network characteristics of attractions in eight popular European tourist cities by photo sharing and GIS spatial statistics[26]. Kang et al. (2018), Lee et al (2018) built the spatial network of attractions in Seoul based on social network technologies of spatial statistics, and evaluated the centrality and spatial structure of attractions[27,10].

  Hwang et al (2006) analyzed the multi-cities travel pattern and network structure of international tourists in the United States, explored the role of cities among tourists in different regions, and proposed to establish a business model of tourist destination bundling[28]. Bhat and Milne (2008) conducted a survey of the embedding, density and centrality of destination marketing network in New Zealand, and found that the centrality of players and the structural characteristics of the overall network affected cooperative marketing[29]. Scott et al (2008) used centrality and other standards as criteria to compare the network structure and network cohesion of four different types and development stages of tourist destinations in Australia, pointed out that the higher degree of industrialization and the larger scale, the stronger cohesion of destinations[30].

  Bianchi et al (2020) adopted the Social Network Analysis (SNA) to study the policy network, focused on the coproduction process of tourist public policies for disables by looking at the network that facilitates communication among the actors taking part in the process[31]. Seok et al (2021) explored how the structure of international tourism has changed longitudinally by using a social network analysis[32].

3.I also call your attention to the illustrations. The tables are difficult to read because some titles are too long and the numbers are easier to read (and compare) when they are aligned to the right and not centred. The same with figures number 1 to 3, which have too much side and inside information (too 'noisy').

The illustrations and tables in the paper were typeset,as follows:

The size of Figure 1 (lines 371-379) and Figure 5 (lines 553-559) has been adjusted to improve the clarity of illustrations.

Resize Figure 2 (lines 403-405), Figure 3 (lines 418-419), and Figure 4 (lines 452-453).

In lines 360-361, Table 1, the title is simplified, and the sorting number identifier (1-31) is deleted. The numbers are right aligned.

In lines 495-498, Table 2 are typeset, the continuation table is used for two pages, and the numbers are right aligned.

4.References

According to the template provided by the editorial department, the reference format was proofread, and checked with the notes in the text one by one.

Note that some references only have issues and no volumes, so only (issues) are marked in the references.

See lines 648-771.

Thank you again for your valuable suggestions on the revision of the paper, which made me learn a lot.

Because there may be some deviation in my understanding of your modification suggestions. If the modification is not in place, please point out what needs to be further modified, and I will try my best to modify it.

Best wishes

Haijian Li

2022.11.20

Reviewer 2 Report

The paper investigates the structural characteristics and evolutionary process of tourism investment spatial correlation 9 network in China by using the revised gravity model and SNA. The manuscript is interesting and overall well written. Below some comments to further raise its quality.

Line 55. Although the Authors argue that “The existing research results show that scholars explore the spatial-temporal pattern 55 and evolutionary process of tourism investment by using geo-statistical methods”, no references to existing works are provided here. I believe that a relevant sentence like this should be duly supported by references.

The Authors should emphasise the manuscript’s novelty. At the moment, the novel contribution of the paper is not clear at all.

Literature review is overall well organised and updated. However, I suggest the Authors to build two or more hypotheses to be tested in the empirical part. In other words all the current empirical results should be conceived as tests to the hypotheses presented in section 2. This may significantly contribute to strengthen the nexus between the literature review and the empirical exercise carried out in the paper.

The area under investigaton should be better presented. A map could be of help to visually identify the 31 provinces analysed. Similarly, it is essential to describe the socio-economic context and present some figures/statistics concerning tourism in the area.

Information provided in Section 3.3.2 should be re-arranged in a table.

What are the policy implication of your work?

Minor revisions

Line 9: […] process of China’s the tourism […].. Please fix.

Lines 49-51 please rephrase

Suggested references:

Bianchi et al. (2020) Accessible tourism in natural park areas: A social network analysis to discard barriers and provide information for people with disabilities. Sustainability

Ma et al.  (2022) “Analysis of spatial patterns and driving factors of provincial tourism demand in China” Scientific Reports

Seok et al. (2021) “A social network analysis of international tourism flow” Quality and Quantity

Author Response

Dear reviewer:

Thank you very much for your affirmation of the paper and your valuable suggestions for revision. According to the modification opinions, we made serious modifications, as follows:

1、Line 55. Although the Authors argue that “The existing research results show that scholars explore the spatial-temporal pattern 55 and evolutionary process of tourism investment by using geo-statistical methods”, no references to existing works are provided here. I believe that a relevant sentence like this should be duly supported by references.

We added relevant references,

see lines 58-59

 (Yan and Chen, 2017; Su et al., 2017a; Liu, 2017)[7-9].

At the same time, the following sentence also adds literature support,

see lines 61-62

(Lee et al., 2018; Wang et al., 2020; Wang and Xia,2018)[10-12]

2、The Authors should emphasise the manuscript’s novelty. At the moment, the novel contribution of the paper is not clear at all.

This really requires serious consideration. Thank you very much for your suggestions.

See lines 64-66

From the perspective of social network, the paper reveals the evolution process of tourism investment spatial correlation network by analyzing the structural characteristics of tourism investment spatial correlation network in China in four time sections.

3、Literature review is overall well organised and updated. However, I suggest the Authors to build two or more hypotheses to be tested in the empirical part. In other words all the current empirical results should be conceived as tests to the hypotheses presented in section 2. This may significantly contribute to strengthen the nexus between the literature review and the empirical exercise carried out in the paper.

Your suggestion is very good, because this paper is not a structural equation or econometric model, so consider putting forward the main research questions in the literature review so as to respond in the empirical part.

See lines 163-167

Through the research of this paper, need to analyze the following questions: (1) What is the structure of tourism investment spatial correlation network in China? What are the characteristics of the overall network and what is the evolution trend? (2) What is the centrality of each node and the change trend? (3) What are the nodes in the core area and the edge area, and analyze the reasons.

4、The area under investigaton should be better presented. A map could be of help to visually identify the 31 provinces analysed. Similarly, it is essential to describe the socio-economic context and present some figures/statistics concerning tourism in the area.

Supplement China's tourism development data in 2016,

see lines 172-180

The number of domestic tourists was 4.44 billion, with a revenue of 3.94 trillion yuan in China In 2016, up 11% and 15.2% respectively over the previous year. The number of inbound tourists reached 138 million, and the international tourism revenue reached 120 billion US dollars, up 3.5% and 5.6% respectively over the previous year. The total revenue of tourism was 4.69 trillion yuan in 2016, up 13.6% year on year. The comprehensive contribution of the national tourism industry to GDP throughout the year was 8.19 trillion yuan, accounting for 11.01% of the total GDP. 28.13 million people were directly employed in tourism, 79.62 million people were directly and indirectly employed in tourism, accounting for 10.26% of the total employed population in China.

5、Information provided in Section 4.3.2 should be re-arranged in a table.

The description of centrality is adjusted accordingly: replace Sichuan and Jiangsu with Shaanxi and Huanan,

See lines 469-473

The betweenness centrality of the top six provinces, including Beijing, Guangdong, Shandong, Liaoning, Shaanxi and Hunan, was above the mean in 2016. The betweenness centrality of Shaanxi, Gansu and Jilin has increased dramatically by a breakthrough from 0, exceeding the average and entering the top ten in 2016.

In the conclusion, replace Sichuan and Jiangsu with Shaanxi and Huanan in line 577, 595 and 599.

In line 19 of the summary, replace Sichuan and Jiangsu with Shaanxi and Huanan.

6、What are the policy implication of your work?

The research results of tourism investment spatial correlation network hope to provide suggestions for the development of regional tourism investment, which are mainly reflected in the following two aspects:

See lines 586-617

(1) The result of the social network analysis of the tourism investment spatial correlation network in China can optimize the network structure of tourism investment and improve the level of regional tourism integration. The attribute data and relationship data of China's tourism investment are basically consistent. In 2016, Guangdong, Beijing, Shanghai, Zhejiang, Jiangsu and Shandong were the top provinces in terms of fixed asset investment and human resource investment in China's tourism industry. According to the social network analysis, the provinces with the highest degree centrality and closeness centrality include Beijing, Shandong, Guangdong, Jiangsu, Zhejiang and Shanghai. The top six provinces in betweenness centrality include Beijing, Guangdong, Shandong, Liaoning, Shaanxi and Hunan. In the process of regional tourism integration, we should pay attention not only to attribute data, but also to the relationship data of the tourism investment spatial correlation network. It is a requisite to give play to the central and intermediary role of provinces, including Beijing, Guangdong, Shandong, Jiangsu, Zhejiang, Shanghai and Liaoning, Shaanxi, Hunan, to optimize the network structure of tourism investment spatial correlation and to realize the mutual promotion of regional tourism integration and network structure. (2) The result of core-periphery analysis of the tourism investment spatial correlation network in China can lessen the tourism investment gap in core-periphery regions by bringing the functions of different provinces into play. Provinces in the core area should give full play to the advantages of capital, talent and enterprise management, strengthen their core position in the spatial connection network of tourism investment, supply high-quality tourism products, and meet the tourism demand inside and outside the region. The marginal provinces should, according to the stage of economic development, consider ecological environment protection and tourism market demand, adjust measures to local conditions, scientifically plan and appropriately invest in tourism to increase the supply of tourism products to promote the high-quality development of the tourism industry. The density of tourism investment spatial correlation between core-periphery areas, which is affected by the level of economic development, population, tourism resources, transportation accessibility and urbanization, shows a downward trend. We advise them to exert the spatial spillover effect of tourism investment in the core areas, to pull the tourism investment of the periphery areas, to realize the sharing of tourism resources and the tourism market and to promote common development.

7、Line 9: […] process of China’s the tourism […].. Please fix.

Amend as follows:

see lines 10-11

The paper uses the revised gravity model to measure the intensity of tourism investment spatial correlation, constructs the spatial correlation matrix of tourism investment, and uses social network method to analyze the structural characteristics and evolutionary process of tourism investment spatial correlation network based on 31 provinces in China from 2000 to 2016.

8、Lines 49-51 please rephrase

Amend as follows:

see lines 49-56

The global financial crisis (2018) and the COVID-19 (2020) have a significant impact on economic development in China, especially tourism development. China has implemented the strategies of expanding domestic demand to promote economic growth, constructing a new development pattern with the domestic cycles as the main body and the domestic and international double cycles promoting each other. In order to give play to the important role of tourism in expanding domestic demand, the tourism investment promotion plan is implemented to stimulate the tourism consumption potential (Zeng and Cheng, 2021)[6].

9、Supplementary References

Relevant literature on tourism flow research using social network method is supplemented,

see lines 113-139

  Pavlovich (2003) explored the evolution and transformation of tourist destination network by taking Waitomo Caves in New Zealand as an example, and discovered that the higher the density, the stronger the cohesiveness of the destination[22]. Stienmetz and Fesenmaier (2015) built a network of attractions in Baltimore to estimate value of attractions and their connection paths[23]. D'Agata et al. (2013) and Asero et al. (2016) built the spatial network of tourist destination in Sicily, and found the mobility of tourists affecting the shape, dimensions and structure of the network[24,25]. Garcia-Palomares et al. (2015) used social network method to assess the spatial network characteristics of attractions in eight popular European tourist cities by photo sharing and GIS spatial statistics[26]. Kang et al. (2018), Lee et al (2018) built the spatial network of attractions in Seoul based on social network technologies of spatial statistics, and evaluated the centrality and spatial structure of attractions[27,10].

  Hwang et al (2006) analyzed the multi-cities travel pattern and network structure of international tourists in the United States, explored the role of cities among tourists in different regions, and proposed to establish a business model of tourist destination bundling[28]. Bhat and Milne (2008) conducted a survey of the embedding, density and centrality of destination marketing network in New Zealand, and found that the centrality of players and the structural characteristics of the overall network affected cooperative marketing[29]. Scott et al (2008) used centrality and other standards as criteria to compare the network structure and network cohesion of four different types and development stages of tourist destinations in Australia, pointed out that the higher degree of industrialization and the larger scale, the stronger cohesion of destinations[30].

  Bianchi et al (2020) adopted the Social Network Analysis (SNA) to study the policy network, focused on the coproduction process of tourist public policies for disables by looking at the network that facilitates communication among the actors taking part in the process[31]. Seok et al (2021) explored how the structure of international tourism has changed longitudinally by using a social network analysis[32].

10、References

According to the template provided by the editorial department, the reference format was proofread, and checked with the notes in the text one by one.

Note that some references only have issues and no volumes, so only (issues) are marked in the references.

See lines 648-771

11、The illustrations and tables in the paper were typeset,as follows:

The size of Figure 1 (lines 371-379) and Figure 5 (lines 553-559) has been adjusted to improve the clarity of illustrations.

Resize Figure 2 (lines 403-405), Figure 3 (lines 418-419), and Figure 4 (lines 452-453).

In lines 360-361, Table 1, the title is simplified, and the sorting number identifier (1-31) is deleted. The numbers are right aligned.

In lines 495-498, Table 2 are typeset, the continuation table is used for two pages, and the numbers are right aligned.

Thank you again for your valuable suggestions on the revision of the paper, which made me learn a lot.

Because there may be some deviation in my understanding of your modification suggestions. If the modification is not in place, please point out what needs to be further modified, and I will try my best to modify it.

Best wishes

Haijian Li

2022.11.20

Round 2

Reviewer 2 Report

Dear Authors, the manuscript looks now definitely improved. Congratulations!